# Field induced spontaneous quasiparticle decay and renormalization of quasiparticle dispersion in a quantum antiferromagnet

Tao Hong[1], Y. Qiu[2], M. Matsumoto[3], D.A. Tennant[1], K. Coester[4], K.P. Schmidt[5], F.F. Awwadi[6], M.M. Turnbull[7], H. Agrawal[8] & A.L. Chernyshev[9]

The notion of a quasiparticle, such as a phonon, a roton or a magnon, is used in modern condensed matter physics to describe an elementary collective excitation. The intrinsic zero-temperature magnon damping in quantum spin systems can be driven by the interaction of the one-magnon states and multi-magnon continuum. However, detailed experimental studies on this quantum many-body effect induced by an applied magnetic field are rare. Here we present a high-resolution neutron scattering study in high fields on an $S = 1/2$ antiferromagnet $C_9H_{18}N_2CuBr_4$. Compared with the non-interacting linear spin–wave theory, our results demonstrate a variety of phenomena including field-induced renormalization of one-magnon dispersion, spontaneous magnon decay observed via intrinsic linewidth broadening, unusual non-Lorentzian two-peak structure in the excitation spectra and a dramatic shift of spectral weight from one-magnon state to the two-magnon continuum.

[1] Quantum Condensed Matter Division, Oak Ridge National Laboratory, Oak Ridge, Tennessee 37831, USA. [2] National Institute of Standards and Technology, Gaithersburg, Maryland 20899, USA. [3] Department of Physics, Shizuoka University, Shizuoka 422-8529, Japan. [4] Lehrstuhl für Theoretische Physik I, TU Dortmund, D-44221 Dortmund, Germany. [5] Lehrstuhl für Theoretische Physik I, Staudtstrasse 7, Universität Erlangen-Nürnberg, D-91058 Erlangen, Germany. [6] Department of Chemistry, The University of Jordan, Amman 11942, Jordan. [7] Carlson School of Chemistry and Biochemistry, Clark University, Worcester, Massachusetts 01610, USA. [8] Instrument and Source Division, Oak Ridge National Laboratory, Oak Ridge, Tennessee 37831, USA. [9] Department of Physics and Astronomy, University of California, Irvine, California 92697, USA. Correspondence and requests for materials should be addressed to T.H. (email: hongt@ornl.gov).

Quasiparticles, first introduced by Landau in Fermi-liquid theory as the excitations for the interacting fermions at low temperatures, have become a fundamental concept in condensed matter physics for an interacting many-body system. Naturally, quasiparticles are assumed to have long, or even infinite intrinsic lifetimes, because of either weak interactions, or the prohibiting energy-momentum conservation for scatterings. However, this picture can break down spectacularly in some rare conditions. Quasiparticle decay was first predicted[1] and then discovered in the excitation spectrum of the superfluid [4]He (refs 2–4), where the phonon-like quasiparticle beyond a threshold momentum decays into two rotons. In magnetism, spontaneous ($T=0$) magnon decays into the two-magnon continuum were observed by inelastic neutron scattering (INS) in zero field in various valence-bond type quantum spin systems including piperazinium hexachlorodicuprate (PHCC)[5], IPA-CuCl$_3$ (ref. 6) and BiCu$_2$PO$_6$ (ref. 7) as well as in some triangular-lattice compounds[8,9]. The mechanism for this magnon instability is the three-magnon scattering process, which is enhanced in the vicinity of the threshold for the decays of one-magnon to two-magnon states[10–12].

By contrast, the phenomenon of the field-induced spontaneous magnon decay in ordered antiferromagnets (AFMs) is much less mature experimentally. Although the key three-magnon coupling term is forbidden for the collinear ground state, it is present in an applied magnetic field when spins are canted along the field direction, that is, the coupling is facilitated via a field-induced spin noncollinearity. To date, spontaneous magnon decay in canted AFMs has been thoroughly studied theoretically[12–18], but there have been very few detailed experimental studies due to the lack of materials with suitable energy scales. The only experimental evidence was reported by Masuda et al. in an $S=5/2$ square-lattice AFM Ba$_2$MnGe$_2$O$_7$ (ref. 19), where the INS spectra become broadened in a rather narrow part of the Brillouin zone (BZ). For the quantum spin-1/2 systems, the magnon decay effect is expected to be much more pronounced, with the analytical[13,14] and numerical studies[15,16] predicting overdamped one-magnon excitations in a large part of the BZ.

Recently, a novel spin-1/2 metal-organic compound (dimethylammonium)(3,5-dimethylpyridinium)CuBr$_4$ (C$_9$H$_{18}$N$_2$CuBr$_4$) (DLCB) was synthesized[20]. Figure 1 shows the molecular two-leg ladder structure of DLCB with the chain direction extending along the crystallographic $b$ axis. At zero field, the inter-ladder coupling is sufficiently strong to drive the system into the ordered phase and the material orders magnetically at $T_N = 1.99(2)$ K coexisting with a spin energy gap due to a small Ising anisotropy[21].

In this paper, we report neutron scattering measurements on DLCB in applied magnetic fields up to 10.8 T and at temperature down to 0.1 K. In finite fields, our study shows strong evidence of the field-induced spontaneous magnon decay manifesting itself by the excitation linewidth broadening and by a dramatic loss of spectral weight, both of which are associated with the three-magnon interactions that lead to magnon spectrum renormalization and to the kinematically allowed one-magnon decays into the two-magnon continuum.

## Results

**Neutron diffraction results in high magnetic fields.** The magnetic structure at zero field is collinear with the staggered spin moments aligned along an easy axis, that is, the $c^*$ axis in the reciprocal lattice space[21]. The size of staggered moment is $\simeq 0.40\mu_B$, much smaller than the $1\mu_B$ expected from free $S=1/2$ ions, due to quantum fluctuations[21]. When a magnetic field is applied perpendicular to the easy axis, the ordered moments

would cant gradually toward the field direction and saturate at $\mu_0 H_s \sim 16$ T. Figure 2 shows the field-dependent neutron scattering peak intensity (inset: the size of the ordered staggered moment $m$) measured at $\mathbf{q} = (0.5, 0.5, -0.5)$ and $T = 0.25$ K. The intensity initially grows due to the gradual suppression of quantum fluctuations with field and reaches a peak value around 6 Tesla. It then decreases with the increase of field because the spin canting angle becomes large. There is no evidence of a phase transition, which confirms this quasiparticle picture. Within the linear spin–wave theory (LSWT), one may expect a similar semiclassical scenario to apply for the spin dynamics.

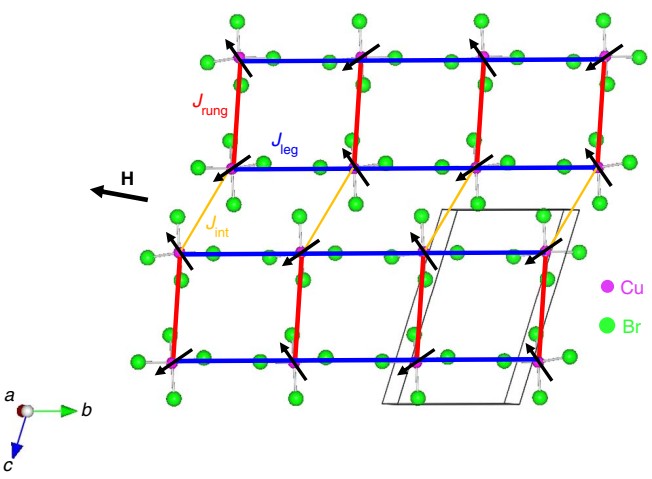

**Figure 1 | Canted spin structure of the coupled ladders in an external magnetic field.** The proposed 2D coupled two-leg ladder structure of DLCB with the ladder chain extending along the $b$ axis. Here, only the CuBr$_4^{-2}$ tetrahedra are shown; other atoms are omitted for clarity. The parallelepiped is the outline of a unit cell. Red, blue and orange lines indicate the intraladder couplings $J_{rung}$, $J_{leg}$ and interladder coupling $J_{int}$, respectively. Black arrows indicate the canted spin structure in an external magnetic field applied along the $[1\bar{1}0]$ direction ($\equiv \hat{\mathbf{x}}$) in the real space.

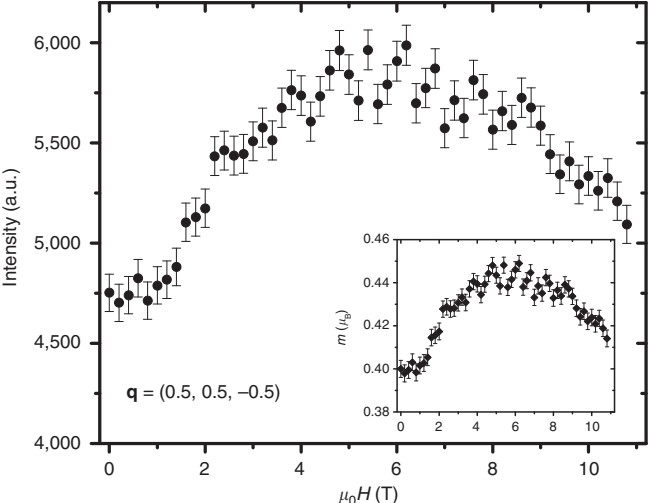

**Figure 2 | Neutron diffraction measurements in an external magnetic field.** Background-subtracted peak intensity as a function of field at $\mathbf{q} = (0.5, 0.5, -0.5)$. Data were collected at $T = 0.25$ K. Inset: the size of the staggered moment as a function of field. All error bars in the figure represent one s.d. determined assuming Poisson statistics.

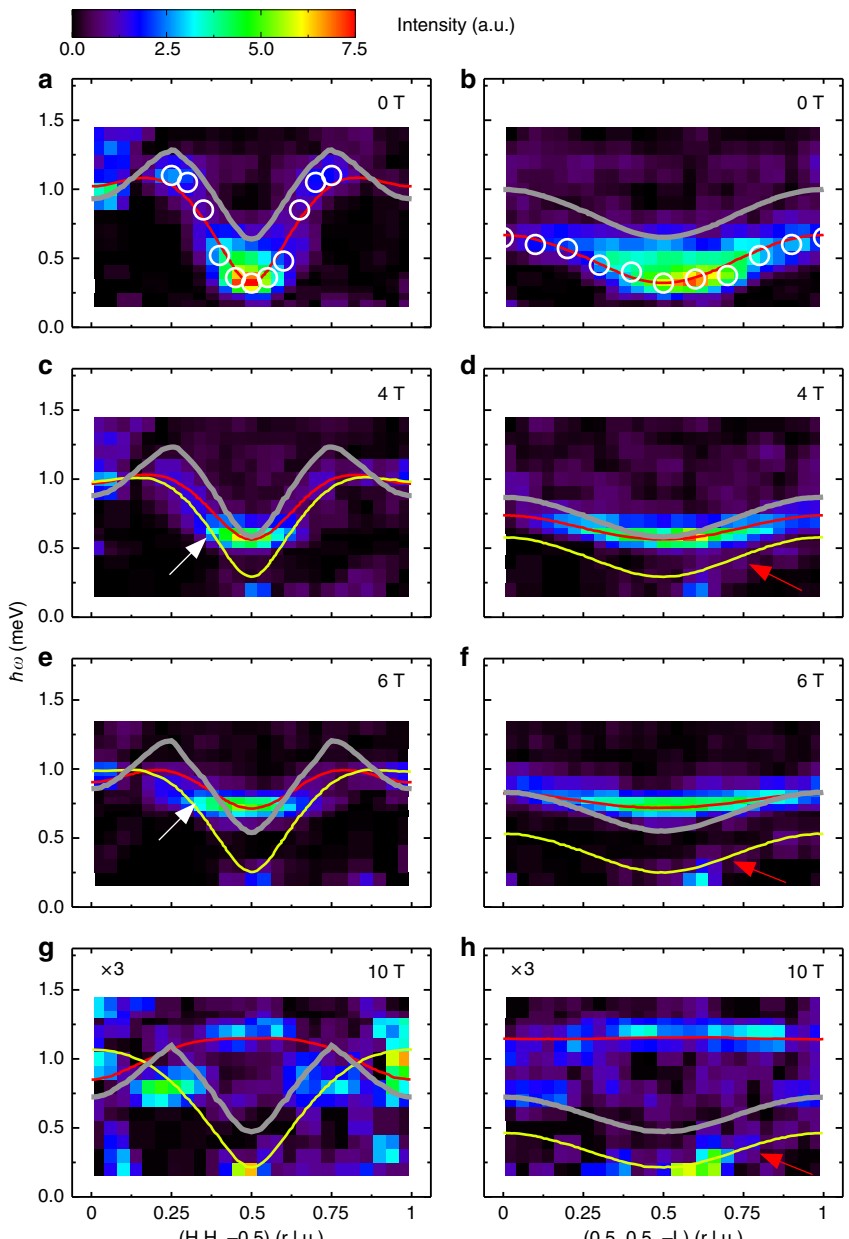

**Figure 3 | INS results in an external magnetic field.** False-colour maps of the excitation spectra as a function of energy and wave-vector transfer along the (H, H, − 0.5) and (0.5, 0.5, − L) directions measured at $T = 0.1\,$K and (**a**,**b**) $\mu_0 H = 0$, (**c**,**d**) $\mu_0 H = 4\,$T, (**e**,**f**) $\mu_0 H = 6\,$T and (**g**,**h**) $\mu_0 H = 10\,$T. Intensity at 10 T was enlarged by a factor of 3 to increase the contrast. Circle points represent the position of maximum intensity at each wave-vector. The white arrows indicate renormalization of the $TM_{high}$ mode. The red arrows indicate the experimental evidence of the $TM_{low}$ mode. Red and yellow lines are the linear spin–wave theory calculations of the acoustic $TM_{high}$ and $TM_{low}$ one-magnon dispersion, respectively. Grey lines are the calculated lower boundary of the two-magnon continuum as described in the Methods section.

**Inelastic neutron scattering results in high magnetic fields.** The spin dynamics, in contrast, undergoes a dramatic change in applied magnetic fields. The corresponding Hamiltonian for a two-dimensional (2D) spin interacting model with nearest neighbour interactions can be written as:

$$\hat{H} = \sum_{\gamma,\langle i,j \rangle} J_\gamma \left[ S_i^z S_j^z + \lambda \left( S_i^x S_j^x + S_i^y S_j^y \right) \right] - g\mu_0 \mu_B H \sum_i S_i^x, \quad (1)$$

where $J_\gamma$ is either the rung, leg or interladder exchange constant— and $i$ and $j$ are the nearest-neighbour lattice sites. $g \simeq 2.12$ is the Landé $g$-factor and $\mu_B$ is the Bohr magneton. The parameter $\lambda$ identifies an interaction anisotropy, with $\lambda = 0$ and 1 being the limiting cases of Ising and Heisenberg interactions, respectively. We assume that $\lambda$ is the same for all three $J'$s in order to minimize the number of fitting parameters to be determined from the experimental dispersion data (Note: this assumption would not affect the main conclusion of the study and is made to prevent overparameterization).

Figure 3 shows false-colour maps of the background-subtracted spin-wave spectra along two high-symmetry directions in the reciprocal lattice space measured at $\mu_0 H = 0$, 4, 6 and 10 T and $T = 0.1\,$K. Spectral weights of the transverse optical branches are weak in the current experimental configurations. Figure 3a,b show the observed spin gapped magnetic excitation spectra at zero field. We employed LSWT for the description of the energy

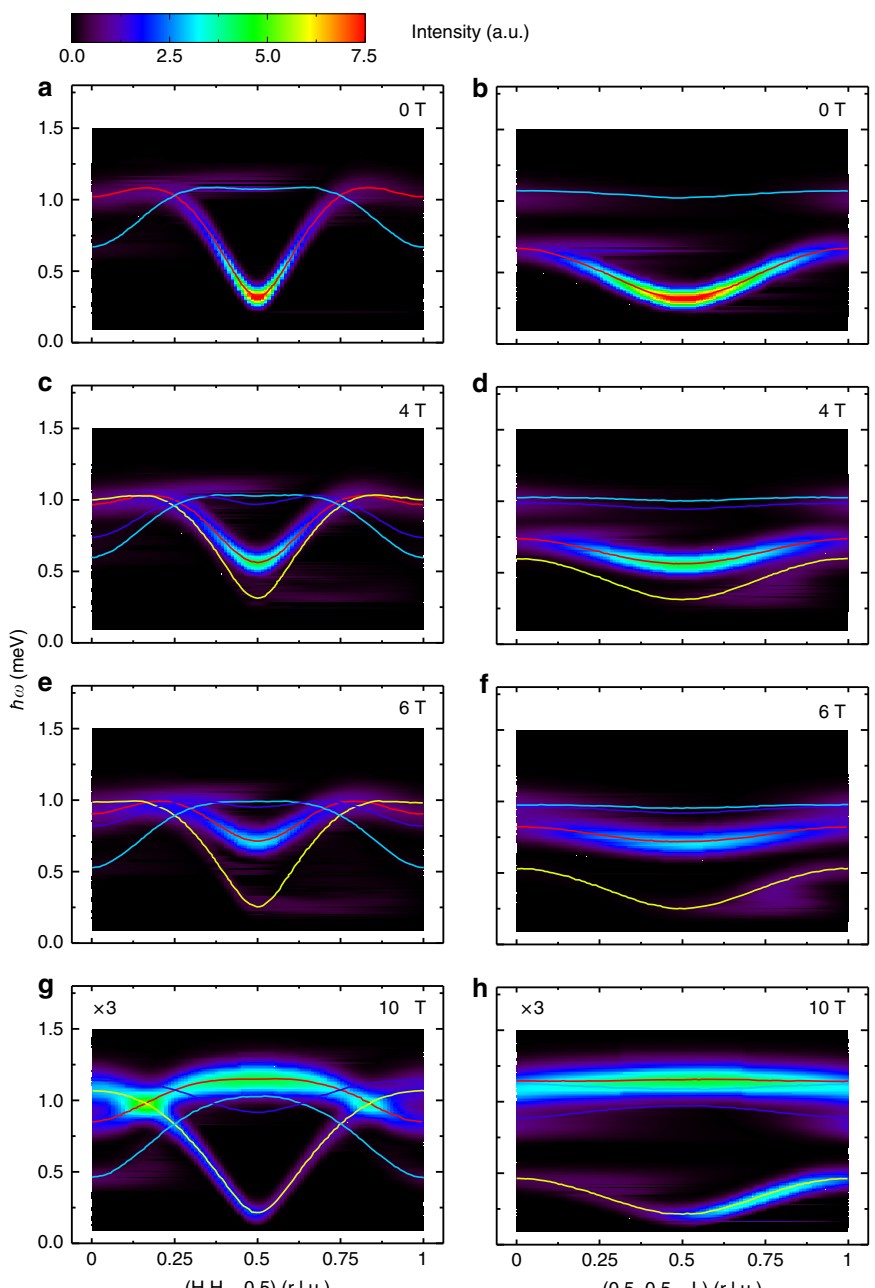

**Figure 4 | Calculations of dynamical structure factors in an external magnetic field.** The linear spin–wave theory calculations of the excitation spectra as a function of energy and wave-vector transfer along the (H, H, − 0.5) and (0.5, 0.5, − L) directions after convolving with the instrumental energy resolution function at (**a,b**) $\mu_0 H = 0$, (**c,d**) $\mu_0 H = 4$ T, (**e,f**) $\mu_0 H = 6$ T and (**g,h**) $\mu_0 H = 10$ T. The spin-Hamiltonian parameters of $J_{leg} = 0.60(4)$ meV, $J_{rung} = 0.70(5)$ meV, $J_{int} = 0.17(2)$ meV and $\lambda = 0.95(2)$ provide the best agreement with experimental data. The calculated intensity at each field was scaled against the total moment sum rule and the same method was also applied for calculations in Fig. 6c. In comparison with experimental data, intensity at 10 T was enlarged by a factor of 3. Red and yellow lines are the calculated acoustic $TM_{high}$ and $TM_{low}$ one-magnon dispersion, respectively. Blue and steelblue lines are the calculated optical $TM_{high}$ and $TM_{low}$ one-magnon dispersion, respectively.

excitations in the Hamiltonian equation (1). The calculated dispersion curves shown as the red lines in Fig. 3a,b using SPINW[22] agree well with the experimental data. The best fit yields the spin-Hamiltonian parameters as $J_{leg} = 0.60(4)$ meV, $J_{rung} = 0.70(5)$ meV, $J_{int} = 0.17(2)$ meV and $\lambda = 0.95(2)$, which are fairly close to the values reported in ref. 21. At $H > 0$, the transverse acoustic mode splits into two branches owing to the broken uniaxial symmetry. The high-energy mode ($TM_{high}$) corresponds to the spin fluctuations along the field direction. The low-energy mode ($TM_{low}$) corresponds to the spin fluctuations perpendicular to both the field direction and the staggered spin

moment ($\equiv \hat{z}$) and is indeed experimentally evidenced in Fig. 3d,f,h (pointed out by the red arrows). Since the neutron scattering probes the components of spin fluctuations perpendicular to the wave-vector transfer, $TM_{low}$ is expected to be weak and is consistent with the LSWT calculations in Fig. 4d,f,h. The dispersion bandwidths of the $TM_{high}$ mode collapse with field, which can also be captured at this LSWT level with the same set of parameters. For instance, the bandwidths along the (H, H, − 0.5) direction at $\mu_0 H = 4$ and 6 T are reduced to 0.44 and 0.25 meV, respectively, from 0.80 meV at zero field. Surprisingly, however, we notice that the $TM_{high}$ mode near the

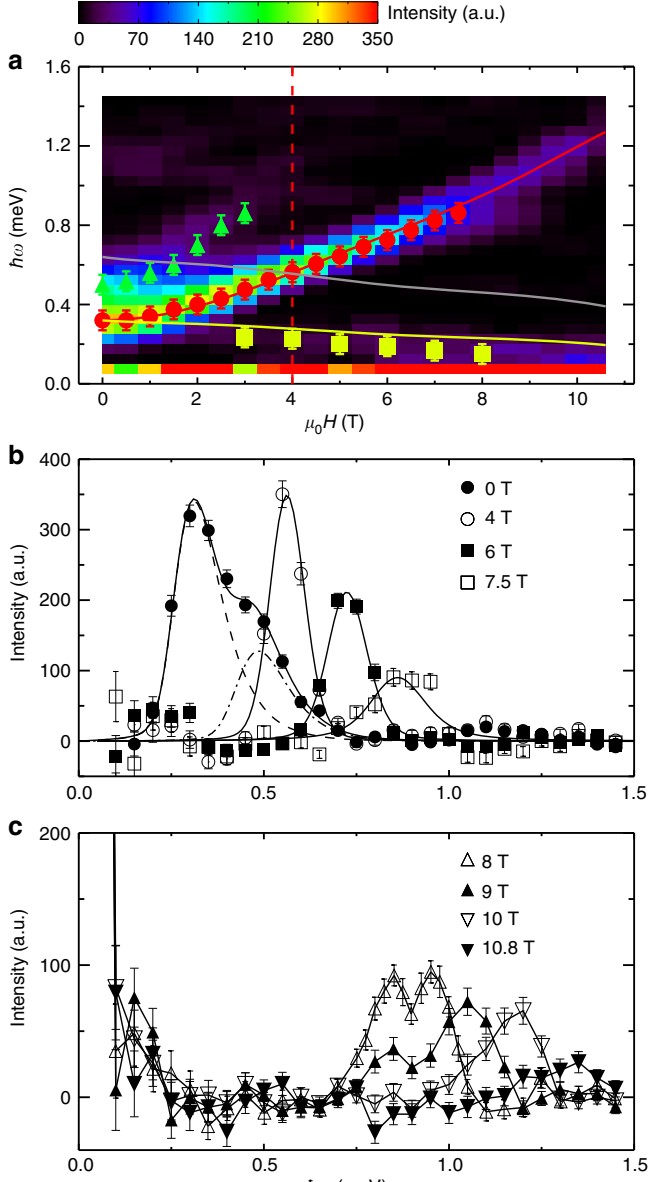

**Figure 5 | Evolution of magnetic excitations with field at the magnetic zone centre.** (**a**) False colour neutron scattering intensity after background subtraction as a function of $\mu_0 H$ and $\omega$ measured at $\mathbf{q} = (0.5, 0.5, -0.5)$ and $T = 0.1$ K. Filled circle (red), square (yellow) and triangle (green) points are peak positions obtained from the fit as described in the text for $TM_{high}$, $TM_{low}$ and LM modes, respectively. Red and yellow lines are calculated by the LSWT. The grey line is the calculated lower boundary of the two-magnon continuum. The dashed line indicates the crossover between the one-magnon state (the red line) and the two-magnon continuum (the grey line) at 4 T; Representative background-subtracted energy scans at (**b**) $\mu_0 H = 0$, 4, 6 and 7.5 T. The solid lines are calculations convolved with the instrumental resolution function. The dashed and dashed dotted lines are calculations for zero field $TM_{high}$ and LM modes, respectively; (**c**) $\mu_0 H = 8$, 9, 10 and 10.8 T. Each line is a guide for the eye. All error bars in the figure represent one s.d. determined assuming Poisson statistics.

BZ centre in Fig. 3c,e visibly bends away from the LSWT calculation in Fig. 4c,e for $\mu_0 H = 4$ and 6 T (pointed out by the white arrows). Such a renormalization of one-magnon dispersion, which is attributed to the strongly repulsive interaction with the two-magnon continuum to avoid the overlap between them, was also recently reported in a different ladder compound, $BiCu_2PO_6$

(ref. 7). Moreover, the spectral weight of one-magnon modes at $\mu_0 H = 10$ T in Fig. 3g,h is much less than what is expected from the LSWT in Fig. 4g,h, indicative of a shift of the spectral weight from one-magon state to multi-magnon continuum.

To investigate this quantum effect in more detail, we have measured excitations at the magnetic zone centre $\mathbf{q} = (0.5, 0.5, -0.5)$ in fields up to 10.8 T and $T = 0.1$ K using a high-flux cold-neutron spectrometer. Figure 5a summarizes the background-subtracted field dependence of the magnetic excitations. Besides the $TM_{high}$ and $TM_{low}$ modes, interestingly, there is also evidence of a mode induced by the spin fluctuations along $\hat{\mathbf{z}}$ but not anticipated by LSWT[23]. At zero field, it is called the longitudinal mode (LM)[24,25], which is predicted to appear in quantum spin systems with a reduced moment size in a vicinity of the quantum critical point[26]. A detailed study of this type of excitation at zero field will be reported elsewhere.

Figure 5b shows the representative background-subtracted energy scans at the magnetic zone centre in different fields. To extract the peak position $\Delta$ and the intrinsic (instrumental resolution corrected) excitation width $\Gamma$, we used the same two-Lorentzian damped harmonic-oscillator model in equation (2) as cross section as used in our previous high-pressure studies[27,28] and numerically convolved it with the instrumental resolution function as described in the Methods section.

$$S(\omega) = \frac{A}{1 - \exp(-\hbar\omega/k_B T)}$$
$$\left[ \frac{\Gamma}{(\hbar\omega - \Delta)^2 + \Gamma^2} - \frac{\Gamma}{(\hbar\omega + \Delta)^2 + \Gamma^2} \right]. \quad (2)$$

The results are plotted in solid (dashed) lines shown in Fig. 5b. The spectral line shape at zero field is the superposition of two such damped harmonic-oscillators and the best fits gives the location of two spin gaps at $\Delta_{TM} = 0.32(3)$ and $\Delta_{LM} = 0.48(3)$ meV, respectively. The LM mode increases with field and is traceable up to 3 T. For the $TM_{high}$ mode, the observed peaks are instrumental resolution limited ($\Gamma \to 0$) up to 4 T although the lineshape looks narrow at 4 T due to the shallow dispersion slope. The steeper slope at zero field induces a broad peak due to the finite instrumental wave-vector resolution. At $\mu_0 H = 6$ and 7.5 T, the line shapes become broadened and the best fits give full-width at half-maximum (FWHM) $2\Gamma = 0.03(1)$ and $0.07(1)$ meV, respectively.

With a further increase of field, the spectral line shape becomes complex. As shown in Fig. 5c, the two-peak structures, distinct from the one-magnon state, appear in the spectrum and are accompanied by a suppression of the magnon intensity. These complex features are consistent with the theoretical prediction for spontaneous magnon decay and spectral weight redistribution from the quasiparticle peak to the two-magnon continuum at high fields[13,18].

## Discussion

Spontaneous magnon decay was also observed in non-collinear transition-metal oxide compounds due to a strong phonon-magnon coupling[29,30]. In DLCB, our consideration of the broadening concerns antiferromagnetic magnons in the proximity of the zone boundary of phonon modes; thus a direct crossing with the long-wavelength acoustic phonon can be excluded. For the metal-organic materials, the optical phonon and spin-wave magnon are usually well-separated, making a phonon branch-crossing scenario unlikely. An entire branch of magnon excitation in DLCB is affected by broadening, making it very hard to reconcile with the phonon-induced scenario where only a select area of the $\mathbf{q}$-$\omega$ space is affected. Moreover, the shift of the magnon energy due to the field would make it exceptionally

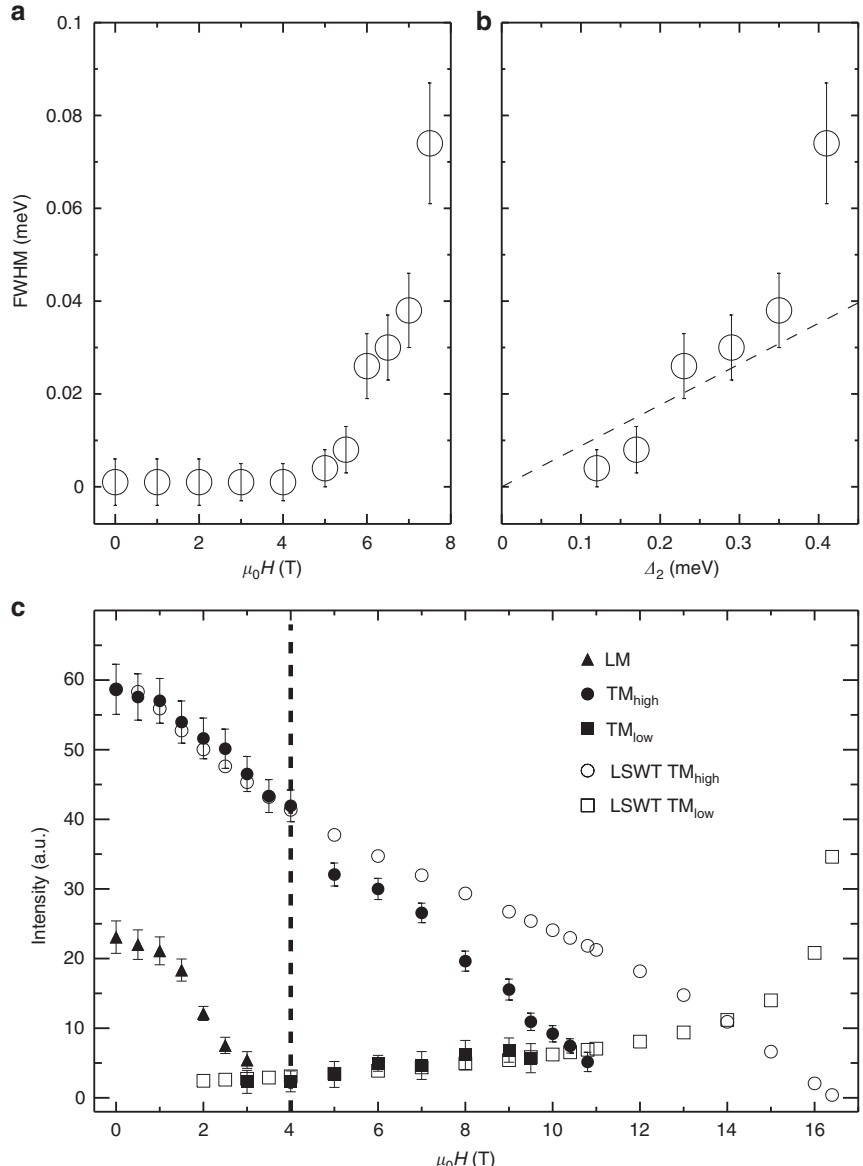

**Figure 6 | Evidences of spontaneous magnon decays at the magnetic zone centre.** (**a**) Field and (**b**) $\Delta_2$ dependence of the intrinsic FWHM of magnon damping. The dashed line is a guide for the eye. (**c**) Field-dependence of the energy-integrated intensity. Filled circle, square and triangle points are the experimental data for the $TM_{high}$, $TM_{low}$ and LM modes, respectively. Open circle and square points are calculations for the $TM_{high}$ and $TM_{low}$ modes, respectively. The dashed line indicates that the spontaneous magnon decay occurs above 4 T. All error bars in the figure represent one s.d. determined assuming Poisson statistics.

unlikely for the resonant-like condition with the phonon branch to be sustained for all the fields. Therefore, the mechanism of the spin-lattice coupling can be ruled out. The observed magnon instability at finite fields in DLCB originates from a hybridization of the single-magnon state with the two-magnon continuum.

The process of one-magnon decays into the two-magnon continuum is allowed if the following kinematic conditions are satisfied:

$$\mathbf{q} = \mathbf{q}_1 + \mathbf{q}_2, \tag{3}$$

$$\varepsilon_2(\mathbf{q}) = \varepsilon_1(\mathbf{q}_1) + \varepsilon_1(\mathbf{q}_2), \tag{4}$$

$$\varepsilon_2(\mathbf{q})^{min} \leq \varepsilon_1(\mathbf{q}) \leq \varepsilon_2(\mathbf{q})^{max}, \tag{5}$$

where $\varepsilon_1$ is the one-magnon dispersion relation, and $\varepsilon_2^{min}$ and $\varepsilon_2^{max}$ are the lower and upper boundary of the two-magnon continuum, respectively.

Since $S_z$ does not commute with the Hamiltonian of equation (1) under the field direction perpendicular to $z$, it is not a good quantum spin number in a field and the two-magnon continuum at finite field can be obtained from any combination between the acoustic and optical $TM_{high}$ and $TM_{low}$ modes. We calculated the lower boundary of the two-magnon continuum, as described in the Methods, plotted as grey lines in Figs 3 and 5a. The upper boundary, which is above the experimental energy range, is not shown. At zero field and along the reciprocal lattice (H, H, −0.5) direction, the lower boundary already crosses with the $TM_{high}$ mode at $H' \approx 0.15$ and $1-H' \approx 0.85$, suggesting that DLCB is prone to magnon decays. Upon an increase of the field, the lower boundary of the two-magnon continuum decreases. At 10 T, the continuum lies underneath the single-magnon branch for the whole BZ as shown in Fig. 3g,h, so the effect of spontaneous decays is expected to be significant.

In the consideration of the magnetic zone centre, the lower boundary of the two-magnon continuum crosses over with the $TM_{high}$ mode at 4 T as shown in Fig. 5a. Figure 6a shows the derived intrinsic FWHM, characteristic of magnon damping, as a function of field up to 7.5 T for the $TM_{high}$ mode. The excitation spectra become even more broadened at higher fields. However, the spectral line shapes become non-Lorenzian so FWHM can not be reliably determined for higher fields. Additional evidence of observation of spontaneous magnon decays is indicated from FWHM versus $\Delta_2$ in Fig. 6b, where $\Delta_2$ is the energy difference between the one-magnon state and the lower boundary of the two-magnon continuum, the quantity of which can be seen as a proxy of the phase space volume for the decay process.

Figure 6c summarizes the field dependence of the energy-integrated intensity of the experimental data plotted with the calculations by LSWT. For comparison purposes, the calculated intensities were scaled by the intensity ratio of $TM_{high}$ at zero field between data and the calculation. Clearly for the $TM_{low}$ mode, data agree well with the calculation. For the $TM_{high}$ mode, data are consistent with the calculation up to $\mu_0 H = 4$ T, above which the intensity drops much faster than the calculation, suggesting the scenario of magnon breakdown. This is consistent with the result shown in Fig. 5a where the $TM_{high}$ mode lies between the lower and upper boundary of the two-magnon continuum above 4 T so the spontaneous magnon decay becomes possible. Due to the maximum field limit accessible for the experimental study, we cannot trace down the critical field where the $TM_{high}$ mode disappears, but its trend points to a much smaller value of such a field than the saturation field $H_s \sim 16$ T, predicted by LSWT. We also notice the similar scenario of dramatic intensity change for the LM mode beyond the crossover with the lower boundary of the two-magnon continuum at 1.5 T (see Fig. 6c), where the crossover with the lower boundary of the two-magnon continuum takes place as shown in Fig. 5a.

With the aid of calculations by the LSWT, our neutron scattering measurements on DLCB show the indication of field-induced magnon decays in the excitation spectra. Direct evidence for the strong repulsion between the one-magon state and the two-magnon continuum is renormalization of the one-magnon dispersion. Our results establish DLCB as the first experimental realization of an ordered $S = 1/2$ AFM that undergoes field-induced spontaneous magnon decays and our study provides much-needed experimental insights to the understanding of these quantum many-body effects in low-dimensional AFMs.

## Methods

**Single crystal growth.** Deuterated single crystals were grown using a solution method[20]. An aqueous solution containing a 1:1:1 ratio of deuterated (DMA)Br, (35DMP)Br, where DMA$^+$ is the dimethylammonium cation and 35DMP$^+$ is the 3,5-dimethylpyridinium cation, and the corresponding copper(II) halide salt was allowed to evaporate for several weeks; a few drops of DBr were added to the solution to avoid hydrolysis of the Cu(II) ion.

**Neutron scattering measurements.** The high-field neutron-diffraction measurements were made on a 0.3 g single crystal with a 0.4° mosaic spread, on a cold triple-axis spectrometer (CTAX) at the HFIR. High-field inelastic neutron-scattering measurements were performed on a disk chopper time-of-flight spectrometer (DCS)[31] (data not shown) and a multi-axis crystal spectrometer (MACS)[32] at the NIST Center for Neutron Research, on two co-aligned single crystals with a total mass of 2.5 g and a 1.0° mosaic spread. At CTAX, the final neutron energy was fixed at 5.0 meV and an 11-T cryomagnet with helium-3 insert was used. At DCS, disk choppers were used to select a 167- Hz pulsed neutron beam with 3.24 meV and a 10-T cryogen-free magnet with dilution fridge insert was used. At MACS, the final neutron energy was fixed at 2.5 meV and an 11-T magnet with dilution refrigerator was used. The background was determined at $T = 15$ K at zero field under the same instrumental configuration and has been subtracted. In all experiments, the sample was oriented in the (H,H,L) reciprocal-lattice plane to access the magnetic zone centre. The magnetic field direction is vertically applied along the $[1\,\bar{1}\,0]$ direction in real space and is thus perpendicular

to the staggered moment direction. Reduction and analysis of the data from DCS and MACS were performed by using the software DAVE[33]. Neutron diffraction at the antiferromagnetic wavevector measures only the staggered moment component of the total spin moment and the size of the staggered moment $m$ is proportional to the square root of the background-subtracted neutron scattering intensity. The field-dependent $m(H)$ at $\mathbf{q} = (0.5, 0.5, -0.5)$ in the inset of Fig. 2 was derived as $m(H) = m(H=0) \times \sqrt{I(H)/I(H=0)}$.

**Convolution with the instrumental resolution function.** In comparison to the observed magnetic intensity, the calculated dynamic spin correlation function $S_\perp(\mathbf{q}, \omega)$ of the spin fluctuation component perpendicular to the wave-vector transfer $\mathbf{q}$ by LSWT was numerically convolved with the instrumental resolutions function as follows:

$$I(\mathbf{q}, \omega) = \int \int d\mathbf{q}' \hbar d\omega' \mathcal{R}_{\mathbf{q}\omega}(\mathbf{q} - \mathbf{q}', \omega - \omega')$$
$$\times \left| \frac{g}{2} F(\mathbf{q}') \right|^2 S_\perp(\mathbf{q}', \omega'), \quad (6)$$

where $F(\mathbf{q})$ is the isotropic magnetic form factor for $Cu^{2+}$ (ref. 34) and $\mathcal{R}_{\mathbf{q}\omega}$ is a unity normalized resolution function that is peaked on the scale of the FWHM resolution for $\mathbf{q} \approx \mathbf{q}'$ and $\hbar\omega \approx \hbar\omega'$ and approximated as a Guassian distribution.

Convolution of the excitation spectra in Fig. 4 was considered as Gaussian broadening of the instrumental energy resolution only. Convolution at the magnetic zone centre in Fig. 5b at zero field was obtained as Gaussian broadening of both instrumental energy and wave-vector resolutions, which were calculated using the Reslib software[35]. For all other fields in Fig. 5b, convolution was considered only as Gaussian broadening of the instrumental energy resolution because the dispersion curve becomes flat near the magnetic zone centre and the spectral linewidth broadening due to the instrumental wave-vector resolution can be safely neglected.

**Determination of the lower boundary of the two-magnon continuum.** There are one acoustic and one optical transverse modes in DLCB at zero magnetic field. When a field is applied perpendicular to the easy axis, either the acoustic or the optical mode splits into two branches ($TM_{high}$ and $TM_{low}$) and it becomes four modes in total. Since $S_z$ is not a good quantum spin number in this case, the two-magnon continuum at finite field was then obtained from any combination between the acoustic and optical $TM_{high}$ and $TM_{low}$ modes.

For the interested $\mathbf{q} = (H,H,L)$, which is equivalent to $(1 + H, 1 + H, L)$, we find the lower boundary of the two-magnon continuum $\varepsilon_2(\mathbf{q})^{min}$ as follows:

$$\varepsilon_2(\mathbf{q})^{min} = \varepsilon_1(\mathbf{q}_1)^{min} + \varepsilon_1(\mathbf{q}_2)^{min}, \quad (7)$$

$$\mathbf{q} = \mathbf{q}_1 + \mathbf{q}_2, \quad (8)$$

where $\varepsilon_1$ is the one-magnon dispersion relation, $\mathbf{q}_1 = (1, 1, 0)$, and $\mathbf{q}_2 = (H, H, L)$. The field dependence of $\varepsilon_1(\mathbf{q}_1)^{min}$ obtained by SPINW in limit of linear spin wave approximation[22] was plotted as the yellow line in Fig. 5a in the main text. The minimum of $\varepsilon_1(\mathbf{q}_2)$ at $\mu_0 H = 0, 4, 6$ and 10 T can be easily deduced from Fig. 4 in the main text. The determined lower boundary of the two-magnon continuum at $\mu_0 H = 0, 4, 6$ and 10 T was then plotted as the grey lines in Figs 3 and 5a.

**Data availability.** The data that support the findings of this study are available from the corresponding author upon reasonable request.

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

## Acknowledgements

T.H. thanks J. Leao for help with cryogenics. A portion of this research used resources at the High Flux Isotope Reactor, a DOE Office of Science User Facility operated by the Oak Ridge National Laboratory. The work at NIST utilized facilities supported by the NSF under Agreement No. DMR-1508249. The work of A.L.C. was supported by the U.S. Department of Energy, Office of Science, Basic Energy Sciences under Award # DE-FG02-04ER46174.

## Author contributions

T.H. conceived the project. F.F.A. and M.M.T. prepared the samples. T.H., H.A. and Y.Q. performed the neutron-scattering measurements. T.H., A.L.C., M.M., D.A.T., K.C. and K.P.S. analysed the data. All authors contributed to the writing of the manuscript.

## Additional information

**Competing interests:** The authors declare no competing financial interests.

**Publisher's note**: 

