## [Peer Review File · Nature Communications]

Reviewer #1 (Remarks to the Author):

The quasiparticle paradigm is an important conceptual building block of modern condensed matter physics. Recent observations of spectacular quasiparticle decays in magnetic systems have illustrated that magnetic quasiparticle system is an excellent model system to study physics of interacting particles. Most of earlier studies have concentrated on materials with singlet ground state with only short-range correlations. Quasiparticles (triplon) in such a system often interact strongly due to anisotropic Dzyaloshinskii-Moriya interactions. In a magnetic system with long range collinear magnetic order, the interaction between magnon quasiparticles is not allowed. Even if small anisotropic interactions do exist, they are probably too small to have any meaningful impact on the excitation spectrum. However, the situation changes when you apply external magnetic field. The moments are canted, and the resulting magnetic structure is no longer collinear. The magnon interaction introduced by the applied magnetic field can lead to magnon decay, which can be studied with inelastic neutron scattering.

The main idea behind the manuscript by Hong and coauthors is that their neutron scattering data shows this field-induced decay behavior of magnons in the organic collinear antiferromagnet DLCB. The authors carried out detailed neutron scattering studies on single crystal samples using both time of flight and triple axis spectrometers. They compared the results with linear spin wave calculation results and ascribed the discrepancy to the magnon decay process. They also found a longitudinal magnon mode which shows magnetic field dependence.

I do believe that the motivation of the study, observation of field-induced quasiparticle decay, is an interesting and important topic. However, overall I do not find the actual data and analysis compelling enough to support the claim, and I do not think the manuscript is suitable for Nature Comm. I have many issues with the manuscript, but some of the more serious ones are:

1)The main results are presented in Fig. 2, which displays field dependence of the magnon dispersion. I am a bit surprised by the quality of data. It is very difficult to actually discern magnon dispersion even for the zero field data (Fig. 2a). The field dependent shift of the zone center intensity is the only unambiguous observation. The dispersion relations (solid lines) more or less go through regions with no intensity. However, the zone center intensity does not seem to change much, except for the 10T data. I am surprised again to see that there is not much intensity observed in this field. If the intensity went into a continuum, that's not clear from the data.

2)Linear spin wave calculation shown in both Figs. 2 and 3 lacks details. Only isotropic exchange couplings are pointed out in Figure 1., but the exchange anisotropy parameter λ was used according to the figure caption of Fig.2. Is this parameter field dependent?

3)Equation 1: Usually one-magnon annihilation cross-section (second term) has $\langle n \rangle$, not $\langle n+1 \rangle$ thermal factor, but the equation seems to suggest that the authors used $\langle n+1 \rangle$ as Bose thermal factor for both terms.

4)Higher field data presented in Fig. 4c is not very convincing. I am not sure one can talk about the lineshape given the statistics. For example in the 8T data, the authors claim two peaks based on a single data point (one low point at 0.9 meV).

5)In order to extract intrinsic width, as presented in Fig. 5, one needs to do a resolution convolution analysis with well-known resolution function. The well-known triple axis spectrometer resolution functions in the past have been calculated assuming more or less parallel beams. Since MACS instrument is not a typical triple-axis instrument, I wonder if the authors considered the effect of focusing optics on the instrumental resolution.

Reviewer #2 (Remarks to the Author):

The manuscript at hand presents experimental data from neutron diffraction and inelastic neutron scattering of a material that is supposed to be a realization of a two-leg ladder antiferromagnet. An analysis of the measured spectra and the broadening of the corresponding peaks in conjunction with a linear spin-wave calculation comes to the conclusion that intrinsic magnon-decay due to magnon-magnon interaction is consistent with the experimental observations. Finally the authors

claim to have evidence for a field-induced magnon decay because the application of a magnetic field opens the phase space for the supposed decay mechanism in this scenario.

The referee agrees that the experimental results are interesting, and the theoretical analysis together with the interpretation is conclusive. Apart from minor comments that are listed at the end of this report, the referee thinks that some parts of the manuscript should be reorganized and made more clear to the reader for the following reasons: The work is mostly experimental, and a theoretical calculation has been included to come to the conclusions mentioned earlier. Therefore, experimental evidences should be more separated from interpretations of results in comparison to theory. More specifically, the referee proposes the following strategy in presenting the central reasoning:

1) For the theoretical calculations, a specific model has been employed. If the referee understands the explanations correctly, the fit parameters have been adjusted to the zero field data. Next the spectra have been calculated for finite field, compared to the experimental data and the phase space for magnon decay has been obtained subsequently. The fitting procedure should be presented first in the analysis.

2) Next the comparison to the spectra at finite field can be presented. (Question: Assuming that the magnetic field enters as a Zeeman term in the calculation? Is the g-factor the same as used for the intensity in Eq. (5)? Why was it chosen accordingly? Is it isotropic?) The conclusion of the authors is now that they find a disagreement between LSW calculations at finite field and their measured data. Can it be ruled out that an extension of the model, for example couplings between the 2D layers (which are not influencing the spectra at zero field), can account for the reduced bandwidth?

3) Finally, the increased broadening of the experimental spectra are explained in terms of quasiparticle decay. In general, there are multiple possibilities that can lead to a broadening or decay. One is the interaction of the modes with the 2 magnon continuum; others might be an interaction with a phonon mode. Can this be ruled out? Has the phonon spectrum been measured, for example with INS at large q ?

In summary, the referee recommends the publication of the manuscript given the presentation of the results is made more clear.

Finally, some comments to improve the manuscript that can be incorporated easily:

a) The spin-wave calculations have been mentioned just in a single sentence. The referee understands that this work is mainly of experimental nature, thus no detailed description of the theory is required. However, the model is just sketched in Fig. 1 and the corresponding parameters are given in the caption of Fig. 3. It would be beneficial to explain how this model has been deduced and whether alternative models have been considered and rejected. More specifically, the numerical value of the "exchange anisotropy parameter" is introduced, but its meaning is not clear to the reader.

b) Fig. 1(a) is not very clear and opens a number of questions: The field direction is given as an arrow. Together with the unusual rotation where 3 of three crystallographic axis are parallel in the projection, it is not easy to understand in which direction the magnetic field points. It is said that it is a projection, but the image seems to be a 3D image since some of the bonds gets shorter. Can you mark the elementary cell, or the magnetic elementary cell? Labels on the bonds for the coupling constants allow the reader to understand the model without rereading the caption again for identification of the color code. The figure shows a classical ground state in a field. Which field value is plotted?

c) In Fig. 1(b) the intensity is plotted as a function of magnetic field together with the staggered moment. How is the staggered moment deduced from the intensity of the neutron peak?

d) Figs. 2,3,4,5: The labels are in a very thin font; especially a font with just black contours instead of filled letters seems not an optimal choice. Especially in Fig. 4(a), the data points with thin black symbols on violet to black background cannot be seen, also the labels on the false color

plot "LM" "TM" will not be readable in print.

e) Fig. 5(b): The quantity Δ_2 is not clearly defined. Is it a purely theoretical quantity, e.g. the difference between the red and gray line in Fig. 4(a) or is it obtained from the experimental spectra at Γ and the theoretical minimum. How can it be seen that the phase space available for magnon decay is proportional to that difference? Is this a result of the explicit solution of the kinematic conditions Eq. (2), (3)? To the referee it seems that the lifetimes in Fig. 5 (a) increase stronger than linear which would be consistent if the phase space also increases stronger than linear.

f) The authors mention that S_z is not a good quantum number without stating the reason. Can you clarify?

g) From Fig. 5(c), the authors conclude that the intensity TM_{high} seem to vanish at lower field experimentally, than the value H_s predicted by LSW. What is the actual value of H_s ? Is it given in terms of the exchange couplings (and g-factor)?

Reviewer #3 (Remarks to the Author):

The paper reports the magnetic field induced breakdown of spin excitations in a quasi one-dimensional (nearly) Heisenberg spin-1/2 ladder. The interest in this work is that the quasi particles that describe the elementary excitations, i.e. the magnons, break down in an applied magnetic field. The work is original, represents an advance on our current experimental knowledge of this process in the lowest spin configuration i.e. spin 1/2, and has the potential to be of interest to those who work in fundamental magnetism. However, there are some points that need to be addressed before I can consider acceptance.

The data presented in Fig. 2 from which the parameters for the Hamiltonian are derived do not seem to be particularly high quality, in that the features that are clearly visible in the color plots are scant: the gap at (0.5,0.5,-0.5) to the TM_{high} mode as a function of applied field, and the band width along (0.5,0.5,-L). However, apart from the flattened dispersion near $H=0.5$ along (H,H,-0.5), the signal from the TM_{high} band is not apparent. Furthermore, there doesn't appear to be any clear signature of the TM_{low} band: I am unconvinced of the presence of features indicated by the red arrows in Figs 2(d) and (f), and nor does there appear to be any signal from these bands in panels (a,c,e,g). The authors have extracted parameters for their Hamiltonian J_{rung} , J_{leg} , J_{int} and λ , but it is far from clear to me how the data permit four independent parameters to be reliably extracted. The authors should give a more convincing analysis. The reason why this is important is because these are the parameters for which features like the crossing of the two-magnon continuum and TM_{high} are computed (Fig. 4(a) and p5, first lines of the paragraph beginning "In the consideration..."): the fact that this occurs at 4 Tesla is an example of the connection between theory and the experiment. To be convinced that there actually is a connection, it is important for the reader to be convinced of the robustness values of the parameters in the model. Similarly, the difference between the linear spin wave theory (LSWT) intensities shown in Fig 5(c) and the experimental values, which therefore indicate spontaneous magnon decay, requires the reader to be convinced that the parameters used to compute the LSWT intensities are reliable. A further point is that the dispersion along (H,H,-0.5) shows marked flattening, as the authors themselves point out (white arrow Fig 2(c) and (e), p3 lines 5-9). If the dispersion is not following LSWT, to what extent can the bare values of J be extracted using LSWT? The authors should include this in their new analysis.

The lifetimes as a function of field (Fig 5(a) and discussion) show a clear effect of damping above 4T. From reading p4 paragraph beginning "Figure 4(b) shows..." and the Methods, these were extracted from the data by convoluting the instrumental resolution function with the spin wave dispersion. It is not clear to me whether this is the dispersion from LSWT with the parameters quoted in the caption to Fig 3, or an empirical dispersion that accounts for the true flatter dispersion around (H,H,-0.5) as highlighted by the white arrows in Fig 2(c) and (e). The authors should explain more clearly, and if from LSWT, what the effect is of using this steeper dispersion that the actual dispersion on their conclusions.

Minor points:

The structure as shown in Fig. 1 is hard to read: I really cannot tell where the a axis is pointing. It looks to me as if the a axis is parallel to b, and the direction of the applied magnetic field is along b (or a), rather than $[1,-1,0]$ as is stated in the caption. A clearer structure more like is presented in Reference 20 would help.

The authors should explicitly give the Hamiltonian in which the parameters J_{rung} , J_{int} etc. appear. Without this, it is not clear what is the significance of the Ising anisotropy, for example.

Fig 3 is very hard to read In Panel (g) because the extent to which there is weak intensity or none at all for some of the modes is obscured by the lines that represent the dispersion.

In Fig 4(a) the markers showing the dispersion of the LM as a function of applied field are essentially invisible above 2T (black on deep purple)

Reply to Reviewer #1

1. The main results are presented in Fig. 2, which displays field dependence of the magnon dispersion. I am a bit surprised by the quality of data. It is very difficult to actually discern magnon dispersion even for the zero field data (Fig. 2a). The field dependent shift of the zone center intensity is the only unambiguous observation. The dispersion relations (solid lines) more or less go through regions with no intensity. However, the zone center intensity does not seem to change much, except for the 10T data. I am surprised again to see that there is not much intensity observed in this field. If the intensity went into a continuum, that's not clear from the data.
We took the referee's criticisms and collected further neutron data using a high-flux cold neutron spectrometer to improve the intensity and signal to noise ratio of the interested magnetic feature in Fig. 2, which makes our conclusions more convincing. Note that it would be difficult to tell the intensity change with field from the false-

color map in Fig. 2. The quantitative comparison and analysis of intensity as a function of field are summarized in Figs. 4(b), (c), and Fig. 5(c).

2. Linear spin wave calculation shown in both Figs. 2 and 3 lacks details. Only isotropic exchange couplings are pointed out in Figure 1., but the exchange anisotropy parameter λ was used according to the figure caption of Fig.2. Is this parameter field dependent?

We include the expression of Hamiltonian and description of the fitting procedure in the updated version of the manuscript. The calculations by the linear spin-wave in Figs. 2 and 3 were done using the software package SPINW (<https://www.psi.ch/spinw/spinw>) as described in the text. For all exchange constants and the interaction-anisotropy parameter λ , we assume that they are field-independent.

3. Equation 1: Usually one-magnon annihilation cross-section (second term) has $\langle n \rangle$, not $\langle n+1 \rangle$ thermal factor, but the equation seems to suggest that the authors used $\langle n+1 \rangle$ as Bose thermal factor for both terms.

We believe the expression in Eq. (1) in the original version of manuscript is correct. $\langle n+1 \rangle$ term comes from the fluctuation-dissipation theorem: $S(\mathbf{q}, \omega) = \langle n+1 \rangle \chi''(\mathbf{q}, \omega)$, where $S(\mathbf{q}, \omega)$ is the dynamic structure factor, $\langle n+1 \rangle = 1/[1 - \exp(-\hbar\omega/k_B T)]$, and $\chi''(\mathbf{q}, \omega)$ is the imaginary part of dynamical susceptibility. Note that this formula applies to both positive and negative values of ω . For the damped harmonic oscillator (DHO) model, $\chi''(\mathbf{q}, \omega) \propto \{\Gamma/[(\hbar\omega - \omega_q)^2 + \Gamma^2] - \Gamma/[(\hbar\omega + \omega_q)^2 + \Gamma^2]\}$ (see “Neutron scattering with a triple-axis spectrometer” by Shirane, Shapiro, and Tranquada and “Principles of condensed matter physics” by Chaikin and Lubensky for more detailed discussions). DHO has been used for accurate modelling of not only phonon and but also magnon damping phenomena (for example, Fåk and Dorner *Physica B* (1997), Stone *et al.*, *Nature* (2006), and Merchant *et al.*, *Nature Physics* (2014)).

4. Higher field data presented in Fig. 4c is not very convincing. I am not sure one can talk about the lineshape given the statistics. For example, in the 8T data, the authors claim two peaks based on a single data point (one low point at 0.9 meV).

We took the criticism and repeated the measurement at 8 T with more data points between the two peaks using the same sample and same instrumental configuration as before. The result clearly shows the unusual double-peak feature.

5. In order to extract intrinsic width, as presented in Fig. 5, one needs to do a resolution convolution analysis with well-known resolution function. The well-known triple axis spectrometer resolution functions in the past have been calculated assuming more or

less parallel beams. Since MACS instrument is not a typical triple-axis instrument, I wonder if the authors considered the effect of focusing optics on the instrumental resolution.

MACS is a new generation cold neutron spectrometer with 20 independent analyzer-detector systems. (Note that data at the magnetic zone center in Fig. 4 were collected using the single detector mode) The large doubly focusing monochromator was used to deliver high intensity neutron beam at sample position. The horizontally focusing mechanism at MACS was designed in such a way that it increases the horizontal acceptance angle, which certainly coarsen the instrumental wave-vector resolution, but not the energy resolution. The effect of the horizontally focusing mechanism of the monochromator can be taken into account by the effective mosaic of the monochromator as discussed in the reference (Nucl. Instr. Meth. Phys. Res. **369**, 169 (1996)). We also point out in the current version that since the dispersion above 4 T becomes flat near the magnetic zone center, the excitation linewidth broadening due to the instrumental wave-vector resolution can be safely ignored and the intrinsic linewidth can be reliably extracted after convolution with the instrumental energy resolution.

Reply to Reviewer #2

1. For the theoretical calculations, a specific model has been employed. If the referee understands the explanations correctly, the fit parameters have been adjusted to the zero field data. Next the spectra have been calculated for finite field, compared to the experimental data and the phase space for magnon decay has been obtained subsequently. The fitting procedure should be presented first in the analysis.
It is correct that the Hamiltonian parameters were derived from the best fit to the excitation spectra at zero field. Following the referee's suggestion, we include a description about the fitting procedure in the updated version of the manuscript.
2. Next the comparison to the spectra at finite field can be presented. (Question: Assuming that the magnetic field enters as a Zeeman term in the calculation? Is the g-factor the same as used for the intensity in Eq. (5)? Why was it chosen accordingly? Is it isotropic?) The conclusion of the authors is now that they find a disagreement between LSW calculations at finite field and their measured data. Can it be ruled out that an extension of the model, for example couplings between the 2D layers (which are not influencing the spectra at zero field), can account for the reduced bandwidth?
The effect to the Hamiltonian with an applied magnetic field is the Zeeman energy term. The anisotropic g-factor for Cu^{2+} typically varies between 2.0 and 2.2. The value of $g=2.1$ used in the paper was determined from the susceptibility measurement [Ref. 20 in the manuscript]. Further evidence for the two-dimensionality in DLCB is provided by the measurements of the inter-layer dispersion. We found that the dispersion is absent within the instrumental resolution. The result is not included here and will be reported elsewhere.

3. Finally, the increased broadening of the experimental spectra are explained in terms of quasiparticle decay. In general, there are multiple possibilities that can lead to a broadening or decay. One is the interaction of the modes with the 2 magnon continuum; others might be an interaction with a phonon mode. Can this be ruled out? Has the phonon spectrum been measured, for example with INS at large q ?

Although we have not measured the phonon spectrum, we can rule out mechanism of the magnon-phonon coupling for the following reasons: (1) Data shown in Figs. 2 and 4 were obtained after subtracting the background, which indicates its magnetic origin; (2) The observed spontaneous magnon decay at the magnetic zone center occurs as the energy of the magnon is changing vs field and happens only if the kinematic condition of Eqs. (3-5) in current version is satisfied. At high fields, the unusual two-peak features, which are also consistent with the theoretical prediction, suggest the weight shift from one-magnon state to two-magnon continuum. Since phonons are not affected by the field, it is very hard to see how, e.g., crossing with a phonon branch would yield the same phenomenology.

- a) The spin-wave calculations have been mentioned just in a single sentence. The referee understands that this work is mainly of experimental nature, thus no detailed description of the theory is required. However, the model is just sketched in Fig. 1 and the corresponding parameters are given in the caption of Fig. 3. It would be beneficial to explain how this model has been deduced and whether alternative models have been considered and rejected. More specifically, the numerical value of the “exchange anisotropy parameter” is introduced, but its meaning is not clear to the reader.

Based on the crystal structure we proposed a minimal two-dimensional (2D) spin interacting model including only the nearest neighbor interactions. We found that this 2D model of coupled spin ladders fully accounts for experimentally observed zero-field magnon dispersions. The detailed information about the choice of the model can be found in our previous paper (PRB 89, 174432 (2014)). Following the comments by the referee, we include the spin Hamiltonian and fitting procedure in the updated version of the manuscript.

- b) Fig. 1(a) is not very clear and opens a number of questions: The field direction is given as an arrow. Together with the unusual rotation where 3 of three crystallographic axis are parallel in the projection, it is not easy to understand in which direction the magnetic field points. It is said that it is a projection, but the image seems to be a 3D image since some of the bonds gets shorter. Can you mark the elementary cell, or the magnetic elementary cell? Labels on the bonds for the coupling constants allow the reader to understand the model without rereading the caption again for identification of the color code. The figure shows a classical ground state in a field. Which field value is plotted?

We replotted Fig. 1(a) as suggested by the referee. The field value is not specified and we just want to demonstrate the spin canted structure when a magnetic field is applied perpendicular to the staggered moment direction.

- c) In Fig. 1(b) the intensity is plotted as a function of magnetic field together with the staggered moment. How is the staggered moment deduced from the intensity of the neutron peak?

When a field is applied perpendicular to the staggered moment direction, spins start canting along the field direction. Since the direction of the staggered moment remains unchanged, the neutron scattering intensity is proportional to square of the staggered moment size, in another word, the field-dependence of the staggered moment size was deduced as $m(H)=m(H=0)*\sqrt{I(H)/I(H=0)}$.

- d) Figs. 2,3,4,5: The labels are in a very thin font; especially a font with just black contours instead of filled letters seems not an optimal choice. Especially in Fig. 4(a), the data points with thin black symbols on violet to black background cannot be seen, also the labels on the false color plot “LM” “TM” will not be readable in print.

As suggested by the referee, we replotted Figs. 2,3,4,5 to improve the figure quality.

- e) Fig. 5(b): The quantity Δ_2 is not clearly defined. Is it a purely theoretical quantity, e.g. the difference between the red and gray line in Fig. 4(a) or is it obtained from the experimental spectra at Γ and the theoretical minimum. How can it be seen that the phase space available for magnon decay is proportional to that difference? Is this a result of the explicit solution of the kinematic conditions Eq. (2), (3)? To the referee it seems that the lifetimes in Fig. 5 (a) increase stronger than linear which would be consistent if the phase space also increases stronger than linear.

Δ_2 is determined from the experimental values for the bottom of the two-magnon continuum, as described in Methods. In brief, we did not intend to claim that the magnon decay rate is linear in Δ_2 for the entire range of Δ_2 , but rather to argue that Δ_2 should serve as a good proxy to a threshold behavior of Γ vs $H-H^*$, where H^* is the threshold field for decays, see also Ref. [17]. To a good approximation, the decay phase space should grow with Δ_2 as the density of states of the decay products grows with the energy of the gapped mode. This proportionality is expected to be strictly linear for the case when the low-energy magnon dispersion is strictly cone-like. One can expect deviations from the linear at larger values of larger Δ_2 when the decay products expand on the flatter parts of the magnon band, which should lead to a stronger than linear dependence. So, in that sense, a correlated growth of Γ and Δ_2 is natural as a trend with the linear dependence being only a guide to the eye. We understand that the information provided in the paper was seriously abbreviated to appreciate all these nuances, so we add a brief clarification to the text.

- f) The authors mention that S_z is not a good quantum number without stating the reason. Can you clarify?

In our experimental setup, the external magnetic field was applied perpendicular to the ordered staggered moment. The z-axis is defined along the staggered moment direction. Therefore, the Hamiltonian is not invariant under a rotation around the z-axis. This means that the angular momentum along the z-axis is not conserved and S_z is not a good quantum number. Simply speaking, it is equivalent to say that the spin operator S_z does not commute with the Hamiltonian.

- g) From Fig. 5(c), the authors conclude that the intensity TM_{high} seem to vanish at lower field experimentally, than the value H_s predicted by LSW. What is the actual value of H_s ? Is it given in terms of the exchange couplings (and g-factor)?

The saturation field H_s is related to the exchange parameters and interaction-anisotropy parameter λ by: $g\mu_B H_s = (1+\lambda)/2 * (2J_{leg} + J_{rung} + J_{int})$. Using the cited values from the paper, H_s is estimated to be 16.4 Tesla, which is fairly close to the experimental value as 16.0(1) Tesla.

Reply to Reviewer #3

1. The data presented in Fig. 2 from which the parameters for the Hamiltonian are derived do not seem to be particularly high quality, in that the features that are clearly visible in the color plots are scant: the gap at (0.5,0.5,-0.5) to the TM_{high} mode as a function of applied field, and the band width along (0.5,0.5,-L). However, apart from the flattened dispersion near $H=0.5$ along $(H,H,-0.5)$, the signal from the TM_{high} band is not apparent. Furthermore, there doesn't appear to be any clear signature of the TM_{low} band: I am unconvinced of the presence of features indicated by the red arrows in Figs 2(d) and (f), and nor does there appear to be any signal from these bands in panels (a,c,e,g). The authors have extracted parameters for their Hamiltonian J_{rung} , J_{leg} , J_{int} and λ , but it is far from clear to me how the data permit four independent parameters to be reliably extracted. The authors should give a more convincing analysis.

After following the comments by the referee, we collected more neutron data using a high-flux cold-neutron spectrometer to improve the data quality of Fig. 2. In the new figure, the TM_{high} mode is clearly visible at 0, 4, and 6 T and also evidenced at 10 T. The TM_{low} mode is also observable in Figs. 2(d), (f), (h), which is consistent with the linear spin-wave calculation as shown in Figs. 3(d), (f), (h). Note that TM_{low} mode reflects the spin fluctuations perpendicular to both the field direction and the staggered spin moment. Its intensity is expected to be weak due to the fact that the neutron scattering probes the components of spin fluctuations perpendicular to the wave-vector transfer.

2. A further point is that the dispersion along (H,H,-0.5) shows marked flattening, as the authors themselves point out (white arrow Fig 2(c) and (e), p3 lines 5-9). If the dispersion is not following LSWT, to what extent can the bare values of J be extracted using LSWT? The authors should include this in their new analysis.

In case the dispersion at high fields bends away from LSWT, J and λ cannot be reliably extracted. In the paper, we assume that the exchange constants J and interaction-anisotropy parameter λ determined from the dispersion at zero-field are field-independent.

3. The lifetimes as a function of field (Fig 5(a) and discussion) show a clear effect of damping above 4T. From reading p4 paragraph beginning “Figure 4(b) shows...” and the Methods, these were extracted from the data by convoluting the instrumental resolution function with the spin wave dispersion. It is not clear to me whether this is the dispersion from LSWT with the parameters quoted in the caption to Fig 3, or an empirical dispersion that accounts for the true flatter dispersion around (H,H,-0.5) as highlighted by the white arrows in Fig 2(c) and (e). The authors should explain more clearly, and if from LSWT, what the effect is of using this steeper dispersion that the actual dispersion on their conclusions.

Since the dispersion curve above 4 T near the magnetic zone center becomes flat, the peak linewidth broadening due to the instrumental wave-vector resolution effect can be safely neglected. It is justified by the observed symmetric spectral line shape at 4, 6, and 7.5 T in Fig. 4(b). In practice, we employed the instrumental energy resolution for the convolution purpose to extract the intrinsic linewidth in Figs. 5(a) and (b).

4. The structure as shown in Fig. 1 is hard to read: I really cannot tell where the a axis is pointing. It looks to me as if the a axis is parallel to b, and the direction of the applied magnetic field is along b (or a), rather than [1,-1,0] as is stated in the caption. A clearer structure more like is presented in Reference 20 would help.

We made the changes as suggested by the referee and updated Fig. 1.

5. The authors should explicitly give the Hamiltonian in which the parameters J_{rung} , J_{int} etc. appear. Without this, it is not clear what is the significance of the Ising anisotropy, for example.

As suggested by the referee, we include the spin Hamiltonian and description of the fitting procedure in the updated version of the manuscript.

6. Fig 3 is very hard to read In Panel (g) because the extent to which there is weak intensity or none at all for some of the modes is obscured by the lines that represent the dispersion. We note that it is generally difficult to tell the intensity change with field from the false-color plots. The quantitative comparison and analysis of intensity as a function of field are summarized in Figs. 4(b), (c), and Fig. 5(c).

7. In Fig 4(a) the markers showing the dispersion of the LM as a function of applied field are essentially invisible above 2T (black on deep purple).

It seems to us that in Fig. 4(a), the LM mode is evidenced at least up to 3 T. As in Fig. 5(c), the LM mode rapidly loses its intensity above 1.5 T, where the lower boundary of two-magnon continuum crosses over with LM mode as shown in Fig. 4(a). Thus, the intensity loss of the LM mode under high fields also shows an evidence of the spontaneous magnon decay.

Summary of major changes from the original version (for convenience, the changes we made are marked with blue color in the text):

1. We are unanimous to add “H. Agrawal” as second to the last author for his great contribution to the sample environment for the neutron scattering experiments.
2. We re-plotted all the figures as commented by the referees.
3. The spin Hamiltonian for a two-dimensional spin interacting model is included in page 1.
4. A paragraph about description of the fitting procedure is included in page 3.
5. We explained why intensity of TM_{low} mode is expected to be weak in the text in page 3.
6. We explained why the phonon-magnon mechanism can be ruled out in the text and cited a paper by Oh, J et al., Nat. Commun. **7**, 13146 (2016) in page 4.
7. We explained why S_z is not a good quantum number in the text in page 4.
8. We explained that the dynamic spin correlation function used for the convolution purpose is the component of spin fluctuations perpendicular to the wave-vector transfer in Methods.
9. We described the detailed procedure of convolution calculation in Figs. 3 and 4 as in Methods.

Reviewer #1 (Remarks to the Author):

I would like to thank the authors for carefully considering my comments and addressing the concerns expressed in detail. I especially appreciate acquiring additional data at two different beamlines to improve the statistics shown in Fig. 2 and Fig. 4. Disappointingly the authors also changed the binning size of the data plotted in Fig. 2, and in that process some information is lost, and I am not sure if the new Figure 2 looks any better than the previous version. My overall assessment remains the same as before. The physics described in the paper is intriguing, but I am still not convinced that the current data and analysis support the main idea of the paper: field-induced quasiparticle decay. As I was reading the new manuscript, a few more questions come to mind:

- 1) The field was applied to (1,-1,0) direction. Given the low symmetry structure, I would imagine the field direction matters. Some explanation about the choice of field direction, and quantitative discussion about the field scale will be useful.
- 2) In Fig. 1b, magnetic scattering intensity is described to increase with field at low field due to the suppression of quantum fluctuation. Can the authors rule out the possibility of intensity change due to the moment direction change?
- 3) Linear spin wave calculation is used throughout the paper, and it is good to see that the authors now include the Hamiltonian used in the calculation explicitly. It looks like they included three exchange interactions along the leg, rung, and interladder directions. However, a universal anisotropy parameter λ was chosen for all three bonds. Since the anisotropy parameter is an effective parameter that will depend on the details of superexchange paths, I do not see a priori reason why this should be identical for all three bonds. More explanation is needed to justify this.
- 4) The decay behavior inside the 2-magnon continuum is puzzling. A dramatic change in the spectrum is observed at much higher field (10T), while nothing much happens when the spectrum crosses the 2-magnon lower bound at lower field. Is there a physical reason for that?

Reviewer #2 (Remarks to the Author):

Having reconsidered the revised manuscript and the reply of the authors to the comments and questions of all referees, it seems that the manuscript has been improved from the previous version: Unclear details were added as requested, and some of the figures are now more clear. Actually, the authors took the concern of the referees seriously and collected further data to demonstrate that the conclusions are robust.

Some comments on the full record:

Given that all 3 referees were not convinced by the presentation of the experimental data and theoretical simulations in Figs. 2 and 3 (asking for more statistics or better presentation), the authors replotted the data. However, the referee thinks that there is still space for improvements in the presentation: Looking at the experimental data, it seems that the data has been rebinned into larger bins, thus reducing the noise level, but at the same time also reducing the resolution in q and ω as evidenced by larger pixels in the revised plots of Fig. 2. Whether or whether not this makes the presentation of the results better cannot be decided. Still, the features in the false color plots are difficult to see: One reason is that the maxima between the 0T case and the 10T case deviate by one order of magnitude. To improve, one could try to stretch the color scale, or plot the log of the data instead to increase the contrast. Also the lines from linear spin-wave theory look very bumpy. Is this a problem of the plotting software? [Note that on some viewers,

Fig. 3 shows artificial vertical or horizontal white stripes on the black areas of the spectra; maybe that is something which can be fixed in the post processing.]

In reply to Comment 3 of Reviewer #2, the authors argue that the signal cannot be of phononic origin because of background subtraction. This argument is not clear to the referee. Excluding phononic excitations should be only possible by either measuring at large q (to take advantage of the expected different prefactors for magnetic and nuclear contributions) or doing a spin-polarized experiment. The second argument brought forward is just a repetition of the argumentation that is already in the manuscript. Also the third argument is not fully conclusive: Indeed, the phonon branches will not be influenced by the magnetic field, but the magnon modes shift and might come close to phonon modes and hybridize and more importantly, the magnon-phonon interaction vertices should be field dependent and could lead to field dependence of the damping by this. Comment a) of Reviewer #2: Now, the discussion is much more accessible without reading the literature and especially Ref. 21. It is worth mentioning that the fit procedure in the present investigation (with different raw data than in Ref. 21, I suppose ?), actually gives very similar results for the parameters in the spin Hamiltonian.

Reviewer #3 (Remarks to the Author):

After reading the revised manuscript and the authors' rebuttals, I am happy that my concerns have been addressed, and can recommend publication in Nature Communications. The interest in the physics was never a question in my view, my reservations centred on the data quality and analysis. Firstly, the new data presented in Fig. 2 improves the believability of the fitted exchange parameters, as the dispersion at zero field along $(H, H, -0.5)$ is now clear. I am a little surprised that the improved data has not resulted in a change to the values of J_{rung} , J_{leg} and λ , which are quoted as having the same values as in the original submission, but this is a minor point and I don't think small alterations to the values would affect any of the later analysis. The authors have also clarified their fitting procedure which now I understand it fully makes my criticism (numbered #2 in their rebuttal) redundant. The explanation and addition in the manuscript about the resolution function convolution in response to #3 and comments by other referees are satisfactory (namely that the flat dispersion at 4T and above means that the only significant contribution comes from the energy resolution). However, I would like to see Fig 5(a) extended to fields, H , for $0T \leq H < 4T$ to show that the intrinsic FWHH is zero within errors, or at least an unambiguous statement to this effect added to the manuscript. This would confirm the robustness of their analysis of the linewidths, and show that the width clearly is turned on when H exceeds the one-magnon/two magnon continuum crossover field at the magnetic zone center. (The data for this already exists of course, as shown in Fig 4a). My other points were all relatively minor, and have been addressed by the authors.

Reply to Reviewer #1

I would like to thank the authors for carefully considering my comments and addressing the concerns expressed in detail. I especially appreciate acquiring additional data at two different beamlines to improve the statistics shown in Fig. 2 and Fig. 4. Disappointingly the authors also changed the binning size of the data plotted in Fig. 2, and in that process some information is lost, and I am not sure if the new Figure 2 looks any better than the previous version.

In the original Fig. 2, the steps of wave-vector and energy transfer for plotting are 0.02 reduced latticed units (r.l.u.) and 0.02 meV, respectively. By contrast, in the revised Fig. 2, the corresponding steps are 0.04 r.l.u. and 0.1 meV (0.05 meV for the gapped TM_{high} mode feature), respectively. The reduction of the wave-vector and energy bin size is mainly due to the choice of two different types of neutron spectrometers. For the time-of-flight spectrometer as DCS, the scattered neutrons were collected by an array of 913 ^3He neutron detectors. In comparison to the Multi-Axis Crystal Spectrometer MACS with 20 ^3He spectroscopic detectors, it is not surprising that data collected at DCS have better instrumental wave-vector resolution than that of MACS. Moreover, due to the different scattering geometry (the incident neutron energy was fixed at

DCS while the final neutron energy was fixed at MACS), the instrumental energy resolution at DCS is also better than that at MACS. During the experiment at MACS, the steps of wave-vector and energy transfer were chosen for not only mapping out the spin-wave dispersions but also the most efficient use of the neutron beam time. We believe that all interesting features can be captured by this method.

Other comments:

- 1) The field was applied to (1,-1,0) direction. Given the low symmetry structure, I would imagine the field direction matters. Some explanation about the choice of field direction, and quantitative discussion about the field scale will be useful.

Since the magnetic propagation wave-vector was determined to be $(1/2 \ 1/2 \ 1/2)$ and the staggered moment direction at zero field is along the c^* axis from Ref. [21], the choice of the scattering plane in the reciprocal-lattice space has to be either (H H L) or (H -H L) in order to make the vertical magnetic-field direction to be perpendicular to the staggered moment direction and access the magnetic zone center. We chose the former case so the vertical direction is $[1 \ -1 \ 0]$ in the real space. Moreover, there is no signature of additional anisotropy perpendicular to the easy-axis c^* , which means that there is a rotational symmetry around the c^* axis. Therefore, physically, the field direction does not matter when it is applied perpendicular to the c^* axis. The related explanation is included in the Methods of the revised manuscript.

- 2) In Fig. 1b, magnetic scattering intensity is described to increase with field at low field due to the suppression of quantum fluctuation. Can the authors rule out the possibility of intensity change due to the moment direction change?

Actually, the ordered moment direction does change with field. As we point out in the “Neutron diffraction measurements in high magnetic fields” part, when an external magnetic-field is applied perpendicular to the staggered moment direction, the ordered magnetic state is changing from the collinear magnetic structure at zero field to the canted antiferromagnetic structure at finite fields, a vector sum of the staggered moment component and a ferromagnetic component along the field direction. Magnitude of ferromagnetic component increases with the increase of field and the staggered moment component would become zero at the saturation field. At the antiferromagnetic wave-vector $q=(0.5 \ 0.5 \ -0.5)$, neutron diffraction measurement with field only sees the staggered moment component. The behavior at low fields in Fig. 1(b) is similar to what was observed in the $S=1/2$ square lattice antiferromagnet (see Fig. 5 in PRB 81, 134409 (2010)) under the similar condition. Moreover, the proposed canted antiferromagnetic structure is supported by the fact the observed spin dynamics at lower fields, especially the TM_{low} mode, can be well reproduced by the LSWT calculation.

- 3) Linear spin wave calculation is used throughout the paper, and it is good to see that the authors now include the Hamiltonian used in the calculation explicitly. It looks like they included three exchange interactions along the leg, rung, and interladder directions. However, a universal anisotropy parameter λ was chosen for all three bonds. Since the anisotropy parameter is an effective parameter that will depend on the details of superexchange paths, I do not see a priori reason why this should be identical for all three bonds. More explanation is needed to justify this.

It is, indeed, true that our choice of the same λ is one possible parametrization of the effects of the easy-axis anisotropy and in reality it can be different for different bonds. The main physical effect of λ is the symmetry breaking, from $SU(2)$ to Ising-like, thus inducing a gap in the goldstone mode. It is, therefore, mostly a parametrization of the gap, which presumably can be achieved by having λ only in one of the J 's, but not the others, as well as other combinations of λ 's for different bonds. Physically, however, as is stated in the revised manuscript, this has been done to limit the number of fitting parameters. Because the anisotropy is quite weak, it is not possible for LSWT calculation to tell apart the λ 's for the three J 's. We thus restrict ourselves with the simplest parametrization and achieve the best LSWT fit with it. It is also justified by the fact that these parametrizations are all identical in their key effect and should lead to quantitatively similar results.

- 4) The decay behavior inside the 2-magnon continuum is puzzling. A dramatic change in the spectrum is observed at much higher field (10T), while nothing much happens when the spectrum crosses the 2-magnon lower bound at lower field. Is there a physical reason for that?

This behavior is in accord with the theoretical result of Refs. [13,17,18] for a simpler model that were also strongly supported by the numerical work [15] and in particular [16], which all have demonstrated a dramatic reconstruction of the spectrum toward the saturation field and well above the first crossing of the two-magnon continuum with the one-magnon branch. This can be understood qualitatively as a kinematic effect, that is, occurring due to the increase of the phase space for magnon decays in higher fields as shown in Fig. 5(b).

Given that all 3 referees were not convinced by the presentation of the experimental data and theoretical simulations in Figs. 2 and 3 (asking for more statistics or better presentation), the authors replotted the data. However, the referee thinks that there is still space for improvements in the presentation:

- 1) Looking at the experimental data, it seems that the data has been rebinned into larger bins, thus reducing the noise level, but at the same time also reducing the resolution in q and ω as evidenced by larger pixels in the revised plots of Fig. 2. Whether or whether not this makes the presentation of the results better cannot be decided.

We did not rebin the data to reduce the noise level and improve the statistics in the revised Fig. 2. The reduction of the wave-vector and energy resolutions is mainly due to the choice of two different types of neutron spectrometers. For the time-of-flight spectrometer as DCS, the scattered neutrons were collected by an array of 913 ^3He detectors. In comparison to the Multi-Axis Crystal Spectrometer MACS with 20 ^3He spectroscopic detectors, it is not surprising that data collected at DCS have better instrumental wave-vector resolution than that from MACS. Moreover, due to the different scattering geometry (the incident neutron energy was fixed at DCS while the final neutron energy was fixed at MACS), the instrumental energy resolution at DCS is also better than that at MACS. During the experiment at MACS, the steps of wave-vector and energy transfer were chosen for not only mapping out the spin-wave dispersions but also the most efficient use of the neutron beam time.

- 2) Still, the features in the false color plots are difficult to see: One reason is that the maxima between the 0T case and the 10T case deviate by one order of magnitude. To improve, one could try to stretch the color scale, or plot the log of the data instead to increase the contrast.

We followed the referee's comment to re-plot the Figs. (2) and (3) using either the logarithmic scale or stretching the color scale. However, neither way works since the intensity at 10 T is weak and quite close to the background level. Instead, we enhanced the intensity at 10 T by a factor of 3 to increase the contrast.

- 3) Also the lines from linear spin-wave theory look very bumpy. Is this a problem of the plotting software? [Note that on some viewers, Fig. 3 shows artificial vertical or horizontal white stripes on the black areas of the spectra; maybe that is something which can be fixed in the post processing.]

The white lines, or “light leaks” are likely a software rendering problem. We have gone to great lengths in testing it with different viewers and do not observe this effect in a majority of them. In Acrobat, Check the Page Display preferences and Unselect “Smooth line art” and “Enhance thin lines” could help see the figures properly. The image we embedded is from a Postscript file which does not contain transparency information. This issue could occur due to Acrobat rendering of transparency of the image.

- 4) In reply to Comment 3 of Reviewer #2, the authors argue that the signal cannot be of phononic origin because of background subtraction. This argument is not clear to the referee. Excluding phononic excitations should be only possible by either measuring at large q (to take advantage of the expected different prefactors for magnetic and nuclear contributions) or doing a spin-polarized experiment. The second argument brought forward is just a repetition of the argumentation that is already in the manuscript. Also the third argument is not fully conclusive: Indeed, the phonon branches will not be influenced by the magnetic field, but the magnon modes shift and might come close to phonon modes and hybridize and more importantly, the magnon-phonon interaction vertices should be field dependent and could lead to field dependence of the damping by this.

We have carefully considered the phonon-induced broadening and have ruled it out. First, our consideration of the broadening concerns antiferromagnetic magnons in the proximity of the corner of the Brillouin zone, thus a direct crossing with the long-wavelength acoustic mode is ruled out and the phonons in question are either optical or zone-boundary ones. Second, for the metal-organic materials, the phonon and magnetic energy scales are usually very well separated. For the energy of the phonons, the family of related polymeric quantum magnets, with similarly small J 's forming different patterns of exchanges, has been recently systematically investigated for their lattice and thermodynamic properties: Phys. Rev. B 92, 134406 (2015) and <https://arxiv.org/abs/1611.06971> The specific heat in them is quantitatively close to the one observed in our material DLCB, see prior work [21], and yields Debye energies ranging from 7 to 14meV, several-fold of the 1meV scale for our magnons, making a phonon branch-crossing scenario extremely unlikely. Lastly, a direct crossing with the phonon branches is well documented in some non-collinear transition-metal oxide compounds, see PRL 99, 266604 (2007) or Ref. [30], where the signatures of such crossings are similar to the resonant-like splitting of the magnon branch that are affecting only select areas of the q -space. In our case, an entire optical branch of magnon excitation is affected by broadening, making it very hard to reconcile with in the phonon-induced scenario. Moreover, the shift of the magnon energy due to field would make it

exceptionally unlikely for the resonant-like condition with the phonon branch to be sustained for all the fields. We include a statement on that in the revised manuscript.

- 5) Now, the discussion is much more accessible without reading the literature and especially Ref. 21. It is worth mentioning that the fit procedure in the present investigation (with different raw data than in Ref. 21, I suppose?), actually gives very similar results for the parameters in the spin Hamiltonian.

We thank the referee for pointing it out. We include a statement on that in the revised manuscript.

Reply to Reviewer #3

After reading the revised manuscript and the authors' rebuttals, I am happy that my concerns have been addressed, and can recommend publication in Nature Communications. The interest in the physics was never a question in my view, my reservations centred on the data quality and analysis. Firstly, the new data presented in Fig. 2 improves the believability of the fitted exchange parameters, as the dispersion at zero field along (H,H,-0.5) is now clear. I am a little surprised that the improved data has not resulted in a change to the values of J_{rung} , J_{leg} and λ , which are quoted as having the same values as in the original submission, but this is a minor point and I don't think small alterations to the values would affect any of the later analysis. The authors have also clarified their fitting procedure which now I understand it fully makes my criticism (numbered #2 in their rebuttal) redundant. The explanation and addition in the manuscript about the resolution function convolution in response to #3 and comments by other referees are satisfactory (namely that the flat dispersion at 4T and above means that the only significant contribution comes from the energy resolution).

We are very pleased that the reviewer is satisfied by our responses to his/her previous comments, and recommends publication in Nature Communications.

- 1) However, I would like to see Fig 5(a) extended to fields, H, for $0 \text{ T} \leq H < 4\text{T}$ to show that the intrinsic FWHH is zero within errors, or at least an unambiguous statement to this effect added to the manuscript. This would confirm the robustness of their analysis of the linewidths, and show that the width clearly is turned on when H exceeds the one-magnon/two magnon continuum crossover field at the magnetic zone center. (The data for this already exists of course, as shown in Fig 4a).

We already have mentioned in the manuscript that "For TM_{high} mode, the observed peaks are instrumental resolution limited ($\Gamma \rightarrow 0$) up to 4 T although the lineshape looks narrow

at 4 T due to the shallow dispersion slope.”. We followed the reviewer’s comment to extend the intrinsic FWHM vs. field for $0 \text{ T} \leq H < 4 \text{ T}$ and updated Fig. 5(a) in the revised manuscript.

Summary of the major changes (for convenience, the changes we made are marked with red color in the text):

1. In Introduction part, added the statement “We assume that is the same for all three J ’s in order to minimize the number of fitting parameters to be determined from the experimental dispersion data”.
2. In Discussion part, added the discussion about the phonon-magnon coupling mechanism “The other possible mechanism, such as the strong phonon-magnon coupling as observed in the non-collinear transition-metal oxide compounds, can be ruled out because: (i) our consideration of the broadening concerns antiferromagnetic magnons in the proximity of the zone boundary of phonon modes, thus a direct crossing with the long-wavelength acoustic phonon can be excluded. (ii) for the metal-organic materials, the optical phonon and spin-wave magnon are usually well-separated, making a phonon branch-crossing scenario unlikely. (iii) in DLCB, an entire branch of magnon excitation is affected by broadening, making it very hard to reconcile with the phonon-induced scenario where only select area of the q - ω space is affected. (iv) the shift of the magnon energy due to the field would make it exceptionally unlikely for the resonant-like condition with the phonon branch to be sustained for all the fields.”.
3. In Methods part under “Neutron scattering measurements.”, added the explanation why the field direction was applied along $[1 \ -1 \ 0]$ in the real space.
4. In Method part, added the statement about “Data availability”.
5. Cited one more reference.
6. Updated Figs. 2, 3, and 5(a) as suggested by the reviewers.

Reviewer #1 (Remarks to the Author):

I would like to thank the authors for responding to my previous comments. However, I am disappointed with some of the responses provided by the authors.

1) About the field direction: The authors did specify the field direction used in the experiments, which is entirely dictated by the experimental condition due to the vertical field magnet. However, my main question was more about the “physical” reason for applying a field in this particular direction. I am not sure how the authors conclude that the system is rotationally symmetric perpendicular to the staggered moment direction (c^* direction). The authors should discuss this with proper reference. Otherwise, they should show direction-dependent magnetization measurements in the supplementary information.

2) I still see several problems with Fig. 1b. The authors quoted another reference to justify the observed behavior at the low field as arising from suppressed quantum fluctuation. However, the experimental system studied in the cited work was a more isotropic system with very small gap compared to the temperature and field scale used in the paper. The current sample has an energy gap of about 0.3 meV, which is much larger than the temperature at which Fig.1b was measured (0.25 K). Although I am somewhat surprised that quantum fluctuation at zero fields is so large even in this case, but I suppose it is not unreasonable. However, the high field behavior, presumably due to the suppressed staggered moment from canting can be quantitatively analyzed using the saturation field value. Also, it's not clear how the absolute staggered moment size was obtained, and why it is proportional to the intensity of magnetic peak rather than the square root of the intensity.

3) I do understand the difficulty with determining the parameters in the Hamiltonian using the linear spin wave theory. (Well, they there should be error bars for these parameters). But since the main argument is based on the observed deviation from the linear spin wave calculation, I would think that the authors should be a little more careful about presenting the effective Hamiltonian used in the calculation. At least they should provide a statement saying that the main conclusion does not depend on the details of the Hamiltonian.

Reviewer #2 (Remarks to the Author):

The referee thanks the authors for answering the remaining concerns about the scientific significance of the results as presented in the manuscript. As also pointed out by other referees, the physical topic is interesting and timely. Nevertheless, one should discuss the results with care to make the claims robust, but not overrate them. In the last revision, the authors have expanded the discussion of why another explanation is unlikely and confirmed this by references to other experimental observations.

In summary, the manuscript has been improved to make many statements from the reply to the referees also clear in the text itself and can be published in its present form.

Reviewer #3 (Remarks to the Author):

I was previously satisfied by the changes that the authors had made to their manuscript, and recommended publication. The further request that I made that lifetimes be added to Fig. 5(a) for fields below 4T has been addressed. The other changes the authors have made in response to the other referees improve the manuscript in my view. The change of intensity scale for the 10T data in fig 2(g) highlights the low statistics of this data, but this is not a point that matters from the overall consistency of the exchange constants - this is one data set out of 8 in the figure.

I recommend publication of the manuscript as it is.

Reply to Reviewer #1

I would like to thank the authors for responding to my previous comments. However, I am disappointed with some of the responses provided by the authors.

1) About the field direction: The authors did specify the field direction used in the experiments, which is entirely dictated by the experimental condition due to the vertical field magnet. However, my main question was more about the “physical” reason for applying a field in this particular direction. I am not sure how the authors conclude that the system is rotationally symmetric perpendicular to the staggered moment direction (c^* direction). The authors should discuss this with proper reference. Otherwise, they should show direction-dependent magnetization measurements in the supplementary information.

With respect to the reason for applying a field in this particular direction, we believed that we explained clearly in the previous reply letter that in order to access the magnetic zone center as $(1/2, 1/2, 1/2)$ and allow the vertical field direction to be perpendicular to the direction of the staggered moment, the sample was aligned in the reciprocal $(H H L)$ plane, which makes the vertical direction $[1 -1 0]$ in the real space.

With respect to the rotational symmetry along the easy axis $c^*(\equiv z)$, if there is an additional anisotropy in the xy -plane, the transverse mode would split into two distinct modes even at zero field (for example, see Fig. 2(a) in PRL 100, 205701 (2008)); there is clearly no evidence for this in the experimental data for DLCB. Therefore, it seems unnecessary to include the field-dependent magnetization measurements in the supplementary information.

2) I still see several problems with Fig. 1b. The authors quoted another reference to justify the observed behavior at the low field as arising from suppressed quantum fluctuation. However, the experimental system studied in the cited work was a more isotropic system with very small gap compared to the temperature and field scale used in the paper. The current sample has an energy gap of about 0.3 meV, which is much larger than the temperature at which Fig. 1b was measured (0.25 K). Although I am somewhat surprised that quantum fluctuation at zero fields is so large even in this case, but I suppose it is not unreasonable. However, the high field behavior, presumably due to the suppressed staggered moment from canting can be quantitatively analyzed using the saturation field value. Also, it's not clear how the absolute staggered moment size was obtained, and why it is proportional to the intensity of magnetic peak rather than the square root of the intensity.

We thank the referee for the comments. However, we disagree with the referee that temperature has nothing to do with the quantum fluctuation. Quantum fluctuations become important for smaller values of the spin quantum number and for lower dimensionality of the interacting spin system. For low dimensional ($D=1$ or $D=2$) systems with the smallest spin value ($1/2$) such as DLCB, quantum fluctuations dominate. One piece of direct evidence is that the size of staggered moment in zero field is about $0.40 \mu_B$, much less than $1 \mu_B$ as expected for free $S=1/2$ ions, due to quantum fluctuations. Therefore, the increase of the size of staggered moment at low fields is indicative of suppression of quantum fluctuations.

With respect to the staggered moment at high fields, as we explained in the previous reply letter, neutron diffraction at antiferromagnetic wavevector measures only the staggered moment component of the total spin moment. Field-dependence of the size of staggered moment in our manuscript was indeed obtained from the square root of the neutron scattering intensity as $m(H)=m(H=0)\times\sqrt{I(H)/I(H=0)}$. Given these observations, we don't think that analysis of the size of the staggered moment using the saturation field value will provide any significant addition to the interpretation of the data.

To address the general concerns raised by the referee, we have included discussions regarding the size of the staggered moment at zero field and have described the procedure for obtaining the size of the staggered moment at finite fields in the revised draft. We also re-plotted the size of the staggered moment as a function of field as an inset of the new Fig. 2 to (hopefully) prevent further confusion.

3) I do understand the difficulty with determining the parameters in the Hamiltonian using the linear spin wave theory. (Well, they there should be error bars for these parameters). But since the main argument is based on the observed deviation from the linear spin wave calculation, I would think that the authors should be a little more careful about presenting the effective Hamiltonian used in the calculation. At least they should provide a statement saying that the main conclusion does not depend on the details of the Hamiltonian.

We thank the referee very much for this suggestion. We have followed the advice of the referee and included the error bars of the Hamiltonian parameters and also a statement stating that the choice of the same λ for all three exchange constants would not affect the main conclusion of our study as part of the revised draft.

Reply to Reviewer #2

The referee thanks the authors for answering the remaining concerns about the scientific significance of the results as presented in the manuscript. As also pointed out by other referees, the physical topic is interesting and timely. Nevertheless, one should discuss the results with care to make the claims robust, but not overrate them. In the last revision, the authors have expanded the discussion of why another explanation is unlikely and confirmed this by references to other experimental observations.

In summary, the manuscript has been improved to make many statements from the reply to the referees also clear in the text itself and can be published in its present form.

We are pleased that the referee is satisfied by our response to his/her previous comments and recommends publication in Nature Communications. We also thank the referee for his/her efforts to improve the presentation of our work.

Reply to Reviewer #3

I was previously satisfied by the changes that the authors had made to their manuscript, and recommended publication. The further request that I made that lifetimes be added to Fig. 5(a) for fields below 4T has been addressed. The other changes the authors have made in response to the other referees improve the manuscript in my view. The change of intensity scale for the 10T data in fig 2(g) highlights the low statistics of this data, but this is not a point that matters from the overall consistency of the exchange constants - this is one data set out of 8 in the figure.

I recommend publication of the manuscript as it is.

We thank the referee for acknowledging that our revised draft satisfactorily addressed his/her concern and those raised by the other referees in the previous reports, as well as for recommending the publication of our manuscript. Again, we appreciate the efforts of the referee to improve the presentation of our results.